# Effect of a national infection control programme in Sweden on prosthetic joint infection incidence following primary total hip arthroplasty: a cohort study

Peter Wildeman [ORCID],[1] Ola Rolfson,[2,3] Per Wretenberg,[1] Jonatan Nåtman,[3] Max Gordon,[4] Bo Söderquist,[5,6] Viktor Lindgren[7]

For numbered affiliations see end of article.

**Correspondence to**
Dr Peter Wildeman;
peter.wildeman@
regionorebrolan.se

## ABSTRACT

**Objectives** Prosthetic joint infection (PJI) is a serious complication following total hip arthroplasty (THA) entailing increased mortality, decreased quality of life and high healthcare costs.

The primary aim was to investigate whether the national project: Prosthesis Related Infections Shall be Stopped (PRISS) reduced PJI incidence after primary THA; the secondary aim was to evaluate other possible benefits of PRISS, such as shorter time to diagnosis.

**Design** Cohort study.

**Setting** In 2009, a nationwide, multidisciplinary infection control programme was launched in Sweden, PRISS, which aimed to reduce the PJI burden by 50%.

**Participants** We obtained data on patients undergoing primary THA from the Swedish Arthroplasty Registry 2012–2014, (n=45 723 patients, 49 946 THAs). Using personal identity numbers, this cohort was matched with the Swedish Prescribed Drug Registry. Medical records of patients with ≥4 weeks' antibiotic consumption were reviewed to verify PJI diagnosis (n=2240, 2569 THAs).

**Results** The cumulative incidence of PJI following the PRISS Project was 1.2% (95% CI 1.1% to 1.3%) as compared with 0.9% (95% CI 0.8% to 1.0%) before. Cox regression models for the PJI incidence post-PRISS indicates there was no statistical significance difference versus pre-PRISS (HR 1.1 (95% CI 0.9 to 1.3)). There was similar time to PJI diagnosis after the PRISS Project 24 vs 23 days (p=0.5).

**Conclusions** Despite the comprehensive nationwide PRISS Project, Swedish PJI incidence was higher after the project and time to diagnosis remained unchanged. Factors contributing to PJI, such as increasing obesity, higher American Society of Anesthesiology class and more fractures as indications, explain the PJI increase among primary THA patients.

## STRENGTHS AND LIMITATIONS OF THIS STUDY

⇒ Long follow-up time to capture prosthetic joint infection (PJI) events and a very high response rate to questionnaires from all units in Sweden operating total hip arthroplasty.
⇒ Used the exact same method for data capturing as a previous study investigating the PJI incidence in Sweden.
⇒ We recognise the limitations of the study, as non-randomised, to establish causality.
⇒ Not all risk factors possibly influencing the outcome were included in the multivariate analysis and a possible underestimation of the PJI incidence due to a lack of ability to identify patients who died in hospital or on long-term intrahospital antibiotic treatment.

## INTRODUCTION

In a world with an increasing elderly population[1] whose members wish to live healthy and mobile lives for most of their lifespans, artificial hip joints improve quality of life (QoL) for millions of patients every year.[2] Excellent long-term pain reduction and improved mobility have popularised the surgical procedure of total hip arthroplasty (THA). However, as with any surgical intervention, THA is associated with procedure-related complications. Aseptic loosening, repeated hip dislocation, thromboembolic event and prosthetic joint infection (PJI) are the most common complications.[3–5] PJIs have multifactorial causes, and extensive research has explored the risk factors for infection.[3] Those associated with PJIs are both endogenous factors such as male gender, obesity, poor diabetic control and active smoking,[6 7] and exogenous factors such as timing of prophylactic antibiotics, surgical site preparation and laminar airflow.[8–10] PJI as a reason for reoperation has gradually increased in recent decades and is now the most common indication for reoperation within 2 years of primary THA.[4] PJIs are associated with increased mortality,[11–13] lower QoL[13 14] and

higher costs for healthcare providers.[15] [16] In an attempt to reduce the PJI burden, the Swedish Orthopedic Association and Swedish National Patient Insurance (SNPI) started in 2009 a nationwide, multidisciplinary infection control programme: PRISS (Prosthetic Related Infections Shall be Stopped). All orthopaedic surgical units (n=81) in Sweden participated in the PRISS Project and were reviewed. The goal of the project was to reduce PJI incidence following total hip and knee arthroplasty in Sweden by 50%. A study by Lindgren *et al*[17] estimated the PJI incidence to 0.9% after primary THA, the years prior to PRISS, 2005–2008. The effect of the PRISS Project on PJI after THA is however unknown. The primary aim of the present study was therefore to investigate if the PRISS Project reduced the incidence of PJI following THA; the secondary aims were to evaluate other possible benefits of the initiative, that is, shorter time to diagnosis, and to identify pathogens involved in PJIs.

## METHODS AND ANALYSIS
### Design
The PRISS Study is a cohort study.

### Registries
All public and private orthopaedic clinics performing THA operations report to the Swedish Arthroplasty Register (SAR). Approximately 98% of all THAs in Sweden are reported to the registry.[4] Details such as indication for surgery, type of implant, surgical approach, body mass index (BMI) and American Society of Anesthesiology (ASA) class are reported. The Swedish Prescribed Drug Registry (SPDR) is operated by the Swedish National Board of Health and Welfare, and all pharmacies in Sweden automatically report complete details regarding dispensed drugs, including type of drug, amount dispensed, instructions from prescribing doctor, and dates of prescription and dispensing.

### SNPI and the PRISS Project
The SNPI is a public insurance company jointly owned by its policy holders, the Swedish regions and county councils. Its primary goals are to insure its owners, handle claims, compensate patients affected by healthcare-associated complications and address patient safety. In 2008, 33% of all claims were from the orthopaedic specialty.[18] In 2009, the PRISS Project was initiated jointly by the SNPI and the Swedish Orthopedic Association.[18] The project started in 2009 and by 2012, all orthopaedic units (n=81) had been enrolled in it. It was an interdisciplinary project for safer THA and total knee arthroplasty (TKA) operations involving several professional organisations such as the Swedish associations for orthopaedic, infectious disease specialists, surgical nursing, orthopaedic nursing, physical therapy and infection control. A self-assessment tool, site inspections and expert group evaluations were performed on all units to evaluate the routines before, during and after arthroplasty surgery.[18] 85% of all implemented measures were improvement in procedures and protocols. Factors such as optimising patients before surgery, basic hygiene routines, optimising antibiotic prophylaxis, improving operating theatres (eg, colony-forming unit measurements, locked rooms and maximum number of persons in operating rooms) and postoperative wound care were among those factors addressed (table 1).

However, the impact of optimisation of these measures in the setting of the PRISS Project, on the PJI risk, has not been explored.

**Table 1** Examples of procedures and improvements addressed in the PRISS Project, 2009–2012

| Preoperative | Operation environment | Postoperative |
|---|---|---|
| No-smoking information, medicines | Improved hygiene routines, standardisation of protocols | Wound treatment, protocol for changing dressing, sterile environment |
| BMI limit | CFU measurement (<5 CFU/m$^3$) | Early detection |
| Diabetic control | Locked operating room | Patient information, sign of infection, contact information |
| Haemoglobin levels | Maximum number of people in operating room | |
| Patient information | | Removing sutures at department |
| Skin-sanitising procedures | Normothermia, regular check of body temperature | Early detection of infection, phone screening 1–2 weeks postop |
| Full body examination | Improved and disposable clothing for all personnel | Follow-up instructions for primary healthcare |
| Clear criteria for cancellation, active infection | WHO checklist for follow-up improvement | |
| Preoperative decolonisation | Antibiotic prophylaxis, type, timing, dose and intervals | |

BMI, body mass index; CFU, colony-forming unit; PRISS, Prosthetic Related Infections Shall be Stopped.

**Table 2** Demographic data for patients undergoing primary total hip arthroplasty, pre-PRISS (2005–2008) and post-PRISS (2012–2014)

| | Pre-PRISS (n=49 259) | Post-PRISS (n=49 946) |
|---|---|---|
| Mean age (SD) | 69 (11) | 69 (11) |
| Female, sex | 50 (29 078/49 259) | 50 (28 645/49 946) |
| BMI ≥30 | 22 (2612/11 969) | 24 (11 227/47 445) |
| ASA classification ≥3 | 17 (2252/13 069) | 20 (9595/48 868) |
| Indication for operation | | |
| Primary OA | 83 (40 884/49 207) | 80 (39 632/49 804) |
| Secondary OA | 0 (22/49 207) | 2 (1183/49 804) |
| Acute trauma, hip fracture | 7 (3348/49 207) | 9 (4325/49 804) |
| Complication trauma | 3 (1413/49 207) | 3 (1715/49 804) |
| Sequelae of childhood hip disease | 2 (1001/49 207) | 2 (936/49 804) |
| Femoral head necrosis | 3 (1235/49 207) | 2 (1162/49 804) |
| Inflammatory joint disease | 2 (1016/49 207) | 1 (562/49 804) |
| Tumour | 1 (288/49 207) | 1 (291/49 804) |
| Surgical approach | | |
| Direct lateral | 44 (21 317/48 889) | 48 (23 836/49 942) |
| Posterior | 55 (26 786/48 889) | 51 (25 654/49 942) |
| Other* | 2 (786/48 889) | 1 (454/49 942) |
| Implant fixation, any cemented† | 89 (43 376/48 761) | 82 (40 709/49 945) |

Data shown as % (n) unless otherwise indicated.
*Trochanteric osteotomy, minimally invasive surgery.
†Cemented, hybrid and reversed hybrid.
ASA, American Society of Anesthesiology; BMI, body mass index; OA, osteoarthritis; PRISS, Prosthetic Related Infections Shall be Stopped.

## Study design, participants and setting

A new method of postoperative infection surveillance introduced by Lindgren *et al* combined national medical registry data with medical record review,[17] determining PJI cumulative incidence within 2 years of primary THA to be 0.9% (95% CI 0.9% to 1.0%) between 2005 and 2008. From that study, we extracted patient demographic data, on patients with a primary THA (all indications) and a PJI diagnosis to compare with the post-PRISS cohort (table 2).

The present study used the same method, including diagnostic criteria for PJI,[19] to determine the incidence of PJI within 2 years of primary THA over the 2012–2014 period. From SAR, we extracted all primary THA procedures (n=49 946 in 45 723 patients) between 1 January 2012 and 31 December 2014 (figure 1). Using personal identity numbers, we matched this cohort with the SPDR and identified 2240 patients undergoing a total of 2569 primary THAs with continuous outpatient consumption of antibiotics for a minimum of 4 weeks minimum of 2 years after each primary THA. For each THA, a questionnaire was sent to the operating unit, where a physician reviewed the medical records of that patient to verify that he/she had been treated for PJI. From the questionnaire, information on date of diagnosis, clinical presentation and surgical treatment was obtained (online supplemental table 1).

99% (n=2531 of 2569) of the questionnaires were returned along with the microbiology results. PJI after revision THA as well as superficial infections were excluded.

## Patient and public involvement

None.

## Statistics

Continuous variables were presented as means (with SDs) and medians (with IQRs), as appropriate. The $\chi^2$ test and Fisher's exact test were used to compare categorical data such as sex, BMI ≥30 and ASA ≥3. The t-test was used to evaluate between-group differences in age. Cumulative incidence of PJI with 95% CIs was estimated using the Kaplan-Meier estimator. HRs with 95% CIs were estimated using Cox regression adjusted for age, sex and diagnosis (osteoarthritis (OA)/other). Patients were followed for 2 years from primary operation until the first PJI and censored for death and revision for other causes than PJI. A sensitivity analysis, in the part of the cohort with complete data, and a Cox regression also adjusted for ASA and BMI were estimated.

All tests were two tailed, and statistical significance was defined as a p value of <0.05 or a 95% CI excluding an OR equal to 1.00. Statistical analyses were performed using SPSS V.25 (IBM Corp).

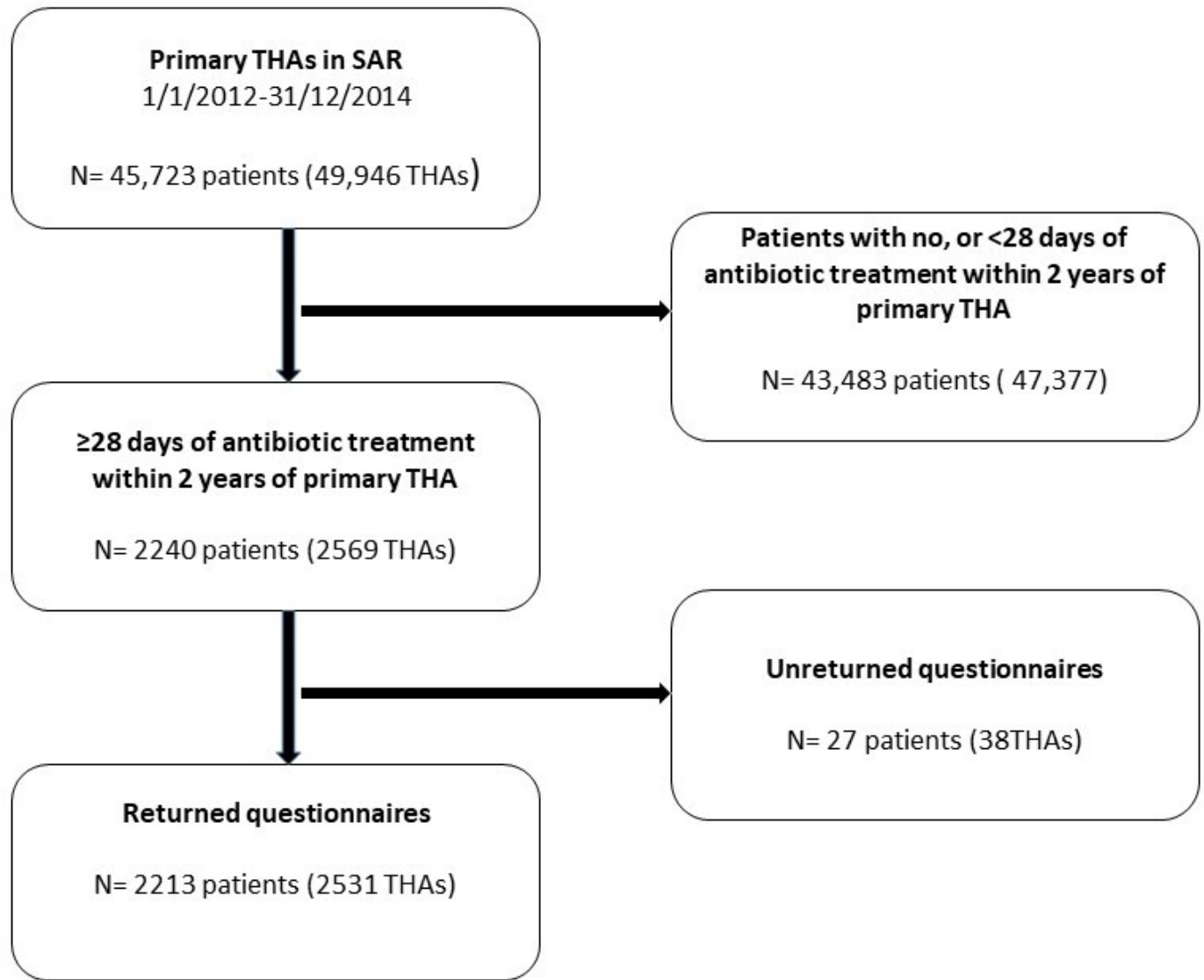

**Figure 1** Study flow chart of primary total hip replacement with antibiotic treatment ≥28 days followed up for review of medical records. SAR, Swedish Arthroplasty Register; THA, total hip arthroplasty.

## RESULTS

580 patients treated for a PJI were identified (online supplemental table 2), resulting in a nationwide 2-year cumulative incidence of 1.2% (95% CI 1.1% to 1.3%) over the 2012–2014 period as compared with 0.9% (95% CI 0.8% to 1.0%) in the 2005–2008 period. There was an increasing trend of PJI incidence during the study period (figure 2) and a significant increase of 30% during the whole study period (p<0.001). However, the multivariate Cox regression model indicated no statistically significant difference in the cumulative incidence pre-PRISS versus post-PRISS PJI (HR 1.1 (95% CI 0.9 to 1.3)).

The highest incidence was noted in university hospitals (1.5%), followed by regional (1.4%) and county and private hospitals (both 0.9%). After controlling for the potential confounders of age, sex, indication for surgery, BMI ≥30 and ASA ≥3, we found no statistically significant differences in PJI incidence among the different hospital classes. For individual orthopaedic units, the cumulative

incidence ranged from 0.0% to 5.9%; in two units, the incidence had decreased and in three, it had increased compared with before the PRISS Project. The median time from primary THA to PJI diagnosis was 23 days (IQR 48) (online supplemental figure 1), not significantly different from the 24 days before the PRISS Project (IQR 46, p=0.51).

In table 3, the distribution of the microbial agents isolated from PJIs pre-PRISS and post-PRISS is presented.

The most common microorganisms, isolated in pure culture or as part of polymicrobial growth, were *Staphylococcus* spp (72% pre vs 68% post, p=0.9). In pure culture, *S. aureus* was the most abundant species, with a minor increase post-PRISS (20% vs 26%, p=0.04), and one isolate was found to be methicillin resistant (ie, methicillin-resistant *S. aureus* (MRSA)) versus none pre-PRISS. Coagulase-negative staphylococci (CoNS) were the second most commonly isolated group of bacteria (21% vs 24%, p=0.2). The number of polymicrobial PJIs,

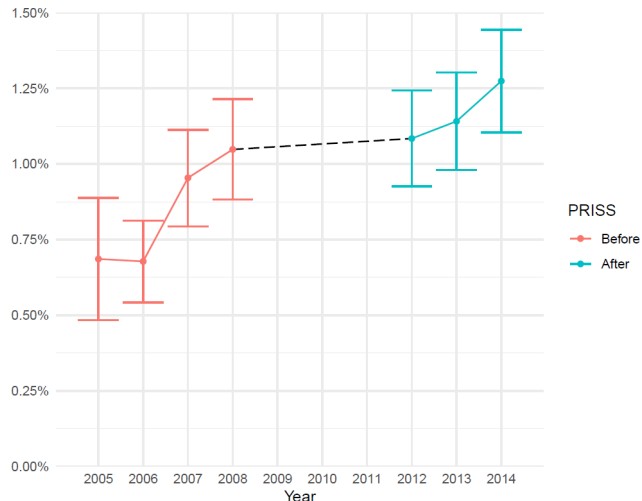

**Figure 2** Trend of cumulative prosthetic joint infection incidence pre-PRISS and post-PRISS, 2005–2014. PRISS, Prosthetic Related Infections Shall be Stopped.

with two to four different bacterial species, did not differ significantly pre-PRISS versus post-PRISS (27% vs 25%, p=0.5).

## DISCUSSION

We investigated the effect of PRISS on PJI incidence in patients with primary THA in the years following the project (ie, 2012–2014) with a follow-up period of 2 years. We also examined time to diagnosis and infecting microorganisms. We found a 30% increase in PJI incidence after

**Table 3** Distribution of microorganisms isolated from prosthetic joint infections (PJIs) of the hip, pre-PRISS (2005–2008) and post-PRISS Project (2012–2014)

| Pathogens | Pre-PRISS (n=442) | Post-PRISS (n=580) |
|---|---|---|
| *Streptococcus* spp | 7% (32) | 8% (44) |
| *Staphylococcus* spp | | |
| *S. aureus* | 20% (89) | 26% (150*) |
| Coagulase-negative staphylococci | 24% (108) | 21% (122) |
| *Enterococcus* spp | 4% (16) | 3% (20) |
| Enterobacterales | 3% (15) | 4% (25) |
| Anaerobes | 1% (3) | 3% (17) |
| Miscellaneous† | 3% (13) | 1% (6) |
| Polymicrobial‡ | 27% (120) | 25% (146) |
| Unknown§ | 10% (46) | 9% (50) |

*Including one methicillin-resistant *S. aureus*.
†For example, *Bacillus, Corynebacterium, Mycobacterium, Pseudomonas*.
‡Polymicrobial findings (two to four organisms). Pre-PRISS 100% of polymicrobial PJIs included *Staphylococcus* spp vs 82% post-PRISS Project.
§Culture-negative PJIs and missing data.
PRISS, Prosthetic Related Infections Shall be Stopped.

as compared with before the PRISS Project, that is, 1.2% vs 0.9%,[17] but there was no statistically significant difference, after controlling for confounders such as BMI, ASA and indication for surgery. Time to diagnosis remained unchanged pre-PRISS versus post-PRISS (ie, median 23 vs 24 days).

Different sources of error may have affected the PJI incidence level reported here. One source of error was that the specific reason for antibiotic prescription could not be established in all cases, with a risk of selection bias For example, if a patient had moved from the county where he/she initially underwent surgery, local medical record review was impossible. Another limitation concerns identifying the patient cohort through SPDR. Since SPDR only handles outpatient prescriptions, patients who died in hospital or were hospitalised during the course of treatment could not be detected, with a risk of introducing immortal time bias. However, the mortality rate is very low following arthroplasty,[20] limiting this effect. According to the Swedish national guidelines,[21] the advocated antibiotic treatment for PJI is 2 weeks of intravenous and 10 weeks of oral antibiotic therapy. Inpatient care for such a long time is rare in Sweden, which also minimises the immortal time bias. In addition, the sensitivity of the 2011 Musculoskeletal Infection Society criteria has been questioned.[22] All these limitations may serve to underestimate the true incidence of PJI. Nevertheless, the study was performed using the exact same methodology as in the prior study by Lindgren *et al*,[17] establishing the PJI incidence before PRISS and having the same limitations in both studies, making the results comparable. The study was non-randomised thus limiting its power to establish causality. The time period for the post-PRISS cohort was selected to be immediately following the completion of the PRISS Project, minimising the regression toward the mean effect. SAR started to collect BMI and ASA data in 2008, limiting the number of patients for whom this information is available. Over the study period, we could identify a trend of increasing comorbidity of the patients (age, BMI, ASA class) and proportion of patients with hip fracture/secondary OA/sequelae after fracture.

As the cause of PJI is multifactorial, any initiative to reduce the infection incidence must be broad and include all healthcare disciplines involved. The PRISS Project included implementations of a bundle of actions to improve preoperative, intraoperative and postoperative care following expert reviews.[18] All recommendations were voluntary for the orthopaedic units, possibly limiting the effect of best-practice methods. It is also possible that there are unknown confounders contributing to an increase in PJI incidence that were not measured.

Most infections were diagnosed in the first 2 weeks following THA and infections were only sporadically identified after the first 90 days (online supplemental figure 1). This implies that most infections originate from the surgical procedure or during the early postoperative period, indicating that properly timing the first dose of prophylactic antibiotics as well as very careful wound care

in the early postoperative period are essential. This also implies that the improved postoperative procedures and protocols of PRISS, such as wound treatment protocols, phone screening and patient information, had limited impact on PJI incidence. There was no change in time to diagnosis compared with before the PRISS Project (ie, 23 vs 24 days). This time to diagnosis is likely acceptable, as infection treatment with debridement, antibiotics and implant retention (DAIR) has a chance of successful outcome (ie, infection eradication) if initiated within 28 days of infection for acute infections.[21 23–25] If a PJI has progressed for more than 4 weeks, extensive surgery, that is, one or two stage exchange is advocated for a successful outcome.

The risk of PJI following the first year of surgery is approximately 1% annually,[3 17] a risk that seemed to be unaffected by the PRISS Project. Late infections include both those contracted during surgery or the early postoperative phase progressing to chronic PJIs, and acute haematogenous infections, which by definition could not be affected by the PRISS Project measures. Although haematogenous infections can be treated successfully with a DAIR procedure within 3 weeks of infection onset,[25] this treatment algorithm is less likely to be successful in a procedure for chronic PJI.[24 26 27] In this study, we have not tried to distinguish between the two types of late infections.

The PJI incidence before PRISS was lower at the beginning of the study period, and there was a trend of increasing incidence at the end of 2008.[17] A trend of increasing PJI incidence could also be detected over the 2012–2014 period, representing a continuation of the increasing incidence noted later in the 2007–2008 period (figure 2). This rising trend could have been because determination to achieve improvement diminished following the end of PRISS. The PRISS Project was a national intervention with broad support from participants from all professional organisations in collaboration with SNPI and all orthopaedic units. Despite broad support and an evidence-based approach, PRISS Project did not reduce PJI incidence. During the 2008–2014 period, the number of patients with severe systemic disease (ASA $\geq$3) increased, possibly contributing to the increasing PJI incidence over this period.[28] More obese patients and patients with hip fractures underwent THA post-PRISS (table 2). Early operation is very important for a patient with a hip fracture,[29] but may also risk non-compliance with preoperative routines, thus contributing to the PJI risk. Thompson *et al* concluded that the PRISS Project did not have any effect on PJI incidence after primary TKA with similar pre-PJI and post-PJI incidence,[30] but did not explore the effect of comorbidities. If the trend of increasing comorbidities applies to the patients with TKA, this could imply that the PRISS Project could have had an effect on the PJI incidence for TKA.[30] However, fracture as indication for primary TKA is rare as compared with THA, limiting this effect on PJI incidence for TKA. Although PJI incidence has progressively increased over the years in Sweden, it is comparable with that reported in an international context.[31 32] Some risk factors for infection cannot be influenced (eg, sex, age and reason for THA), whereas some, although modifiable (eg, heart failure, diabetes and obesity),

are hard to influence especially if the patient is in pain and has difficulty walking before THA, meaning that they were largely unaffected by the preoperative measures of the PRISS Initiative. However, it is possible that the PJI incidence might have been even higher without the PRISS Project and its various benefits, such as more patients receiving correct initial treatment and therefore being cured of infection.[3 30]

Staphylococci were by far the most common aetiological agent of PJI, in agreement with international reports[33] and national observations before PRISS.[17 34] Since only one MRSA case was found, the vast majority of *S. aureus* were susceptible to most antibiotics. However, on the contrary, many of the CoNS were not determined to species level, and *S. epidermidis*, the most common staphylococcal species among CoNS causing PJI, are in contrast to *S. aureus* often multidrug resistant.[35] Whether a pattern of increasing multidrug resistance could explain the increasing PJI incidence was not evaluated here.

## CONCLUSIONS

The PRISS Project did not reduce the PJI incidence and the time to diagnosis remained unchanged. This study indicates a lack of impact of non-mandatory recommended measures, despite being evidence based, to be implemented to reduce multifactorial complications such as PJI of the hip when the incidence is low. Higher proportions of obese, comorbid and fracture patients likely explain the minor increase in PJI incidence, and it is possible that in modern healthcare, endogenous risk factors might affect the PJI incidence more than exogenous (and modifiable) risk factors. Further studies of patients with a high risk of PJI, such as fracture, revision surgery and comorbid patients, are warranted to reduce overall incidence of PJI and its severe complications. An increased focus on PJIs may have contributed to a better treatment outcome, but this needs further studies.

**Author affiliations**
[1]Department of Orthopedics, Faculty of Medicine and Health, Orebro University, Orebro, Sweden
[2]Department of Orthopaedics, Institute of Clinical Sciences, Sahlgrenska Academy, University of Gothenburg, Gothenburg, Sweden
[3]Swedish Arthroplasty Register, Registercentrum Vastra Gotaland, Gothenburg, Sweden
[4]Department of Clinical Sciences, Danderyd Hospital, Karolinska Institutet, Stockholm, Sweden
[5]School of Medical Sciences, Faculty of Medicine and Health, Örebro University, Orebro, Sweden
[6]Department of Infectious Diseases, Faculty of Medicine and Health, Orebro University, Orebro, Sweden
[7]Department of Molecular Medicine and Surgery, Karolinska Institutet, Stockholm, Sweden

**Correction notice** This article has been corrected since it was published. The corresponding email address has been updated. Licence updated to CC BY on 1st August 2024.

**Acknowledgements** The authors would like to thank all orthopaedic units and SAR contact physicians for review of the medical records. The authors would also like to thank Ann-Britt Wall, Department of Orthopedics Region Örebro County, for administrative support.

**Contributors** PWi, OR, PWr, MG, BS and VL designed the study. VL did project supervision and was guarantor of the study. PWi, MG and JN analysed the data.

PWi wrote the manuscript with contributions from OR, PWr, JN, MG, BS and VL. All authors approved the final version of the manuscript.

**Funding** The funding organisations, Research Committee of Örebro University, Region Örebro County (grant no: OLL-917121), Sweden and Patientförsäkringen LÖF, Sweden (grant no: NA) provided financial support for the study.

**Disclaimer** The funders had no role in study design, data collection and analysis, decision to publish or preparation of the manuscript.

**Competing interests** None declared.

**Patient and public involvement** Patients and/or the public were not involved in the design, or conduct, or reporting, or dissemination plans of this research.

**Patient consent for publication** Not applicable.

**Ethics approval** This study involves human participants and was approved by the Regional Ethical Review Board of Gothenburg, Sweden (reference number: 2017/329-17). Participants gave informed consent to participate in the study before taking part.

**Provenance and peer review** Not commissioned; externally peer reviewed.

**Data availability statement** All data relevant to the study are included in the article or uploaded as supplemental information.

**ORCID iD**
Peter Wildeman http://orcid.org/0000-0001-7840-8979

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
