## [Reviewer comments · BMJ Open]

ARTICLE DETAILS

TITLE (PROVISIONAL)	The effect of a national infection control program in Sweden on prosthetic joint infection incidence following primary total hip arthroplasty: a cohort study.
AUTHORS	Wildeman, Peter; Rolfson, Ola; Wretenberg, Per; Nåtman, Jonatan; Gordon, Max; Söderquist, Bo; Lindgren, Viktor

VERSION 1 – REVIEW

REVIEWER	Lemaignen, Adrien Centre Hospitalier Régional Universitaire de Tours, Service de Médecine Interne et Maladies Infectieuses
REVIEW RETURNED	20-Aug-2023

GENERAL COMMENTS	I read with interest the manuscript entitled "The effect of a national infection control program in Sweden on prosthetic joint infection incidence following primary total hip arthroplasty" by P Wildeman et al. Using the exhaustive national registry of THA combined with the Swedish Drug Registry, authors have evaluated the effect on prosthetic joint infections' incidence of a national program aiming to control modifiable risk factors to reduce infection incidence and burden, by comparing pre- and post-program incidences. They found an increasing tendency in incidence of PJI, although not significant after adjustment with risk factors. They explain this finding mostly by a rise of non-modifiable risk factors for PJI in the study population. Although the method used is intrinsically biased in the post-intervention incidence estimation, it is the same than used by Lindgren et al, and therefore a comparison is permitted. Furthermore, this point is well discussed in the limitations. The evaluation of national programs is important to optimize public fundings and interventions and with this in mind, this evaluation deserve publication. However, some minor point could be developed to improve the manuscript. Minor points: Methods: Why did the authors choose the 2012-2014 period ? Would have been possible to get more updated data ? Indeed, with a follow-up of 2 years before censoring, one could have imagined evaluating a more recent cohort of patients, including before COVID-crisis Discussion: -In a Public Health perspective and cost-utility reflexion, authors
--

	could have reported or estimated the annual costs of the PRISS program and discussed its utility. Do the authors think that this program should continue in this form ? Are there improvements to advice with these findings ? -I. 222 : perhaps authors could have specified that "conservative" treatment has a chance of successful outcome when performed before 28 days. After that timing, one- or two-stage prosthesis exchange have quite good outcomes as curative treatment.
--	--

REVIEWER	Adili, Anthony McMaster University Faculty of Health Sciences, Surgery
REVIEW RETURNED	10-Oct-2023

GENERAL COMMENTS	This is an excellent administrative database study looking at a national infection control program and its effects on prosthetic joint infection in hip arthroplasty. This is a very important topic as PJI is uncommon but can be devastating for patients and very challenging for surgeons to treat. I have a few comments:  1. Per RECORD reporting guidelines, the title should specify the study design, e.g. retrospective administrative database study 2. The strengths and limitations summary section only contains strengths. Should also list limitations particularly with this being a non-randomized study that is asserting causality. 3. I'm not sure if it makes sense to combine all THA together, e.g. fractures, tumour cases combined with elective THA for osteoarthritis. This would be more focused if only elective THA was included, or if results were presented separately for the different subgroups by type of surgery. The methods should also explicitly state whether the study includes all types of THA or only the more routine elective THA. 4. Very impressive response rate from surgeons. The authors should be commended. 5. The authors should be very clear about what is adjusted vs unadjusted results. From the unadjusted results it seems that the incidence increased but adjusted looks like it did not, although more details about the adjusted results are needed. With an observational (non-randomized) study it is essential to be very careful and explicit about this. 6. I would like to see a table of the results of the Cox PH regression with the variables adjusted for. 7. I'm not sure that all important confounders were adjusted for. Particularly, diabetes is a very large risk factor for infection. 8. Was the regression adjusted for elective vs emergency THA? 9. The discussion section states "The PJI incidence before PRISS was lower at the beginning of the study period, and there was a trend of increasing incidence at the end of 2008". The authors can't conclude that the incidence was lower before PRISS without a significant p value. The first results paragraph states it was not statistically significant (HR 1.1, 95% CI 0.9-1.3). There are a few other instances of the authors stating that something is lower or higher but there is no statistical test or confidence interval to demonstrate that is it significantly lower or higher.
---

VERSION 1 – AUTHOR RESPONSE

Reviewer: 1

Minor points:

Methods:

Why did the authors choose the 2012-2014 period ? Would have been possible to get more updated data ? Indeed, with a follow-up of 2 years before censoring, one could have imagined evaluating a more recent cohort of patients, including before COVID-crisis.

Thank you for an excellent question. The rationale to choose the 2012-14 period were as follows: the PRISS project started in 2009 and ended in the beginning of 2012 (95% of all centers were included before 2012). To reduce the risk of loss of adherence to the PRISS protocol we selected the period directly adjacent to PRISS project. The time/number of patients were also equivalent to the pre PRISS study by Lindgren et al.

Discussion:

-In a Public Health perspective and cost-utility reflexion, authors could have reported or estimated the annual costs of the PRISS program and discussed its utility.

The cost –utility analysis was beyond the scope of this study and the exact direct and non-direct cost are not fully known. Patientförsäkringen LÖF (the Swedish National Patient Insurance direct costs were approximately 12 million Swedish crowns (approximately 2 million dollars in todays value. 72 Swedish orthopedic department that performed prosthetic surgery participated, health professionals from all parts of the respective department participated in both local, regional and national meetings. These costs have not been estimatedbut are likely many times the cost LÖF had for the project.

Do the authors think that this program should continue in this form ? Are there improvements to advice with these findings?

We don't have data in this study that supports a national driven project equivalent to the PRISS project . However, we believe that every unit should work with infection preventive measures. We don't believe the PRISS project should continue in its current forms since it didn't contribute to a lower PJI incidence.

-I. 222 : perhaps authors could have specified that "conservative" treatment has a chance of successfull outcome when performed before 28 days. After that timing, one- or two-stage prosthesis exchange have quite good outcomes as curative treatment.

Thank you for observing this unclear wording. We have added a clarifying sentence.Please seeP14 Line 252-254

Reviewer: 2

Dr. Anthony Adili, McMaster University Faculty of Health Sciences

Comments to the Author:

This is an excellent administrative database study looking at a national infection control program and its effects on prosthetic joint infection in hip arthroplasty. This is a very important topic as PJI is uncommon but can be devastating for patients and very challenging for surgeons to treat. I have a few comments:

1. Per RECORD reporting guidelines, the title should specify the study design, e.g. retrospective administrative database study

We have changed the title to comply with the reporting guidelines to “The effect of a national infection control program in Sweden on prosthetic joint infection incidence following primary total hip arthroplasty: a cohort study”.

2. The strengths and limitations summary section only contains strengths. Should also list limitations particularly with this being a non-randomized study that is asserting causality.

The “Strengths and limitations of this study” has been updated with limitations. Please see p 4 line 57-64.

3. I'm not sure if it makes sense to combine all THA together, e.g. fractures, tumour cases combined with elective THA for osteoarthritis. This would be more focused if only elective THA was included, or if results were presented separately for the different subgroups by type of surgery. The methods should also explicitly state whether the study includes all types of THA or only the more routine elective THA.

In order to be able to compare our data with Lindgren et al's data we included all diagnosis. In the multivariate analysis we included indication for operation e.g. osteoarthritis/other to compensate for this valid argument. The manuscript has been updated. Please see p9 line 161-162.

4. Very impressive response rate from surgeons. The authors should be commended.

Thank you for your very kind words!

5. The authors should be very clear about what is adjusted vs unadjusted results. From the unadjusted results it seems that the incidence increased but adjusted looks like it did not, although more details about the adjusted results are needed. With an observational (non-randomized) study it is essential to be very careful and explicit about this.

We agree with this statement. We found a 30% increase in the PJI incidence ($p < 0.001$) but it was non-significant after adjustment for relevant confounders. Please see p 10 line 172-174.

6. I would like to see a table of the results of the Cox PH regression with the variables adjusted for your convenience.”

We have attached the documentation for the COX regression for your convenience. Please see file “PRISS Cumulative incidence and Cox regression”.

7. I'm not sure that all important confounders were adjusted for. Particularly, diabetes is a very large risk factor for infection.

We agree that diabetes is a risk factor for infection. We did not have full access of all patient variables in the registries, but selected the available ASA variable as a surrogate variable for patient health status.

8. Was the regression adjusted for elective vs emergency THA?

The regression model included “indication for surgery”. In the analysis we dichotomized indication for surgery to: (Primary OA/Other). Please see table 2 and p 9 Line 161-162.

9. The discussion section states "The PJI incidence before PRISS was lower at the beginning of the study period, and there was a trend of increasing incidence at the end of 2008". The authors can't conclude that the incidence was lower before PRISS without a significant p value. The first results paragraph states it was not statistically significant (HR 1.1, 95% CI 0.9-1.3). There are a few other instances of the authors stating that something is lower or higher but there is no statistical test or confidence interval to demonstrate that is it significantly lower or higher.

Thank you for your comment. That statement refers to data in the study by Lindgren et al (2014) Figure 2 “Cumulative incidence...”. The p-value for the trend test was P=0.004. The pre vs. post post PJI incidence was 30% higher (p<0.001). We have changed the text in the Results section please see p10 line 172-174 and Discussion section p.12 line 209-212

VERSION 2 – REVIEW

REVIEWER	Adili, Anthony McMaster University Faculty of Health Sciences, Surgery
REVIEW RETURNED	13-Feb-2024
GENERAL COMMENTS	The authors have addressed all my previous comments. No further concerns.